# The Effect of Sowing Date on the Biomass Production of Non-Traditional and Commonly Used Intercrops from the *Brassicaceae* Family

**DOI:** 10.3390/plants14233654

**Published:** 2025-11-30

**Authors:** Václav Brant, Andrea Rychlá, Kateřina Hamouzová, Viktor Vrbovský, Pavel Procházka, Josef Chára, Jiří Holejšovský, Theresa Piskáčková, Soham Bhattacharya, Jiří Dreksler

**Affiliations:** 1Faculty of Agrobiology, Food and Natural Resources, Czech University of Life Sciences Prague (CZU), Kamýcká 129, 165 00 Praha-Suchdol, Czech Republic; brant@af.czu.cz (V.B.); pavelprochazka@af.czu.cz (P.P.); chara@af.czu.cz (J.C.); holejsovskyj@af.czu.cz (J.H.); tpiskackova@af.czu.cz (T.P.); bhattacharya@af.czu.cz (S.B.); dreksler@af.czu.cz (J.D.); 2OSEVA PRO Ltd., Purkyňova 10, 764 01 Opava, Czech Republic; rychla@oseva.cz (A.R.); vrbovsky@oseva.cz (V.V.)

**Keywords:** aboveground biomass production, carbon fixation, leaf–stem ratio, non-traditional *Brassicaceae*, root-to-shoot ratio, sowing date optimization

## Abstract

Catch crops play a vital role in agricultural systems by contributing to biomass production, nutrient retention, and carbon sequestration. Among these, species from the *Brassicaceae* family are particularly valuable due to their rapid biomass accumulation, biofumigant properties, and adaptability to diverse environmental conditions. This study presents the first systematic evaluation of biomass characteristics for six non-traditional *Brassicaceae* species under Central European conditions, alongside commonly cultivated representatives of the family. Field experiments were conducted in Eastern Bohemia from 2021 to 2023 to assess biomass production in nine *Brassicaceae* species. Four sowing dates were evaluated, with plant sampling determining aboveground and underground biomass. The results revealed significant species-specific differences in biomass accumulation. *Sinapis alba* and *Raphanus sativus* exhibited the highest biomass, while *Brassica napus* and *Crambe abyssinica* had the lowest. A positive correlation between aboveground and underground biomass was observed across species, though root-to-shoot ratios varied, influencing carbon allocation patterns and soil organic matter inputs. Overall, the results demonstrate that sowing date is a critical factor influencing growth dynamics and reproductive development in these underutilized *Brassicaceae* species. By identifying optimal planting windows, this study contributes to improved management strategies aimed at maximizing biomass production while supporting sustainable cropping practices, including enhanced soil organic carbon stabilization and reduced nutrient losses.

## 1. Introduction

Catch crops have a notable impact within agricultural systems, offering advantages like biomass generation and retention of nutrients. Catch crops and cover crops are plant species grown primarily to protect and improve the soil between main crop cycles. Catch crops are usually sown to capture residual nutrients and reduce leaching, while cover crops provide soil cover to prevent erosion, enhance soil structure and contribute organic matter and carbon sequestration. The biomass yield from catch crops can fluctuate due to variations in species, sowing methods, and environmental elements like soil composition and climatic conditions [1,2,3]. The *Brassicaceae* family is renowned primarily for its ability to act as a biofumigant, aiding in the suppression of crop pests when incorporated into the soil. Brassicas are primarily renowned for their biofumigant properties but are also known for their biomass production [4,5]. Isothiocyanate compounds possess the capability to exhibit toxicity towards weeds [6], nematodes [7], insects [8], and diseases [9,10]. Species of the *Brassicaceae* family are also known for their rapid growth and biomass production, including the utilization of solar energy [3,11]. This biomass production is dependent on climatic and soil conditions [12]. Molinuevo-Salces et al. (2014) and Słomka and Oliveira (2021) observed that *Brassica* species are considered suitable for methane production [13,14]. In the case of *Brassica* cover crops, Heuermann et al. (2019) [15] demonstrated the presence of *S. alba* roots at a soil depth of 0.72 m. Gentsch et al. (2020) [16] pointed out that they influence microbial communities in the lower parts of the soil profile. Although drought and soil salinity tolerance of *Crambe abyssinica* has been documented in the study of Samarappuli et al. (2020) [17], its performance under Central European conditions remains unreported. Similarly, *Eruca sativa* has shown promising biomass potential in South Europe, but data under central European climatic and soil conditions are lacking. These gaps highlight the need for systematic evaluation of non-traditional *Brassicaceae* species in this region.

Research has demonstrated that certain *Brassica* species exhibit increased nitrogen storage underground when accounting for rhizodeposits in catch crops [18]. Pawłowski et al. (2021) [19] consider that *Brassicaceae* species have a notable role in CO_2_ sequestration. In their findings, they stated that sequestration of CO_2_ in the above-ground biomass was observed (6.56 t CO_2_ per ha) for *Sinapis alba*. Catch crops have the capacity to absorb an additional 4 to 6 tons of CO_2_ per hectare per year [20]. Kwiatkowski et al. (2023) [20] employed a carbon content calculation for biomass, assuming a dry weight of 42%. Brant et al. (2022) [21] demonstrated that the carbon content in the dry aboveground and underground biomass of *S. alba* was 42.0% and 42.2%, respectively, while in *Camelina sativa*, it was 39.0% and 44.0%, respectively. The biomass of brassicaceous cover crops exhibits a narrow C:N ratio, which reaches an average value of 18:1 for *S. alba* and *R. sativus* [3]. According to Słomka and Oliveira (2021) [14], the C:N ratio in biomass is also influenced by the location. Their results demonstrated that in one location, the C:N ratio for *S. alba* grown as a cover crop was 15.5:1, while in another, it was 19.9:1. Kolbe et al. (2011) [22] found that the C:N ratio is also dependent on the level of N fertilization and simultaneously pointed out the different C:N ratio in leaves and stems. In *Sinapis alba*, the C:N ratio in leaves ranged from 12–15:1, while in stems, it ranged from 41–55:1. The inclusion of this issue in the literature review is motivated by the need to assess the contribution of individual plant organs, which, according to published data, may differ in their carbon-to-nitrogen (C:N) ratio. The C:N ratio subsequently influences the rate of cover crop biomass biodegradation when left on the soil surface or incorporated into the soil [23].

The production of cover crop biomass generally depends on soil conditions, weather, cultivation region, soil tillage, date of sowing, presence of weeds, etc. [11,24,25,26]. The timing of sowing influences biomass production significantly as well. Spring-sown *S. alba* biomass ranged from <0.5 to 4 t/ha. Early fall biomass ranged from 3 to 5.5 t/ha, and was related to growing degree days (GDD) according to a logistic function [27]. Słomka and Oliveira (2021) [14] reported that the aboveground dry biomass of *S. alba* varied between 0.6 and 0.8 t/ha. The average yield of catch crops’ air-dried biomass (t/ha) achieved in the soil and climate conditions of central Lubelskie Voivodeship was 4.26 t/ha for *S. alba* and 3.40 t/ha for *B. napus* (spring form), according to Pawłowska et al. (2019) [28]. Entrup and Oehmichen (2000) [29] described the negative impact of late sowing of cover crops on aboveground biomass production. Literature provides primarily information regarding aboveground biomass production of brassicaceous cover crop species such as *B. napus*, *S. alba*, and *R. sativus*. However, data for non-traditional brassicaceous species evaluated in the current study, particularly under Central European conditions, remain scarce or unavailable.

Cover crops play a crucial role in soil characteristics and underground biomass production. Walter (1962) [30] highlighted the positive influence of vegetation duration and soil rooting depth. Nonetheless, data regarding underground biomass production from typical cover crops, including *Brassicaceae* species, are limited and uncertain [31]. Data regarding underground biomass production of roots are comparably limited due to variations in evaluation methods, sowing dates, site influences, and other factors. Nonetheless, understanding underground biomass production is crucial for utilizing cover crops as green manure or for mitigating nutrient leaching [32,33,34]. Underground biomass production has the potential to serve as a significant source of carbon sequestration in the soil [35] and could potentially contribute more to the stability of soil organic carbon compared to reincorporating the aboveground residue as green manure [36]. Smucker (1984) [37] and Taylor (1986) [38] pointed out that ignorance of these facts limits the quantification of root biomass production and carbon storage in the soil in relation to field conditions. Estimates of underground plant biomass frequently rely on a fixed allometric relationship between the biomass of aboveground and underground portions of plants. However, environmental and management factors may affect this allometric relationship, making such estimates uncertain and biased [39]. Available information on underground biomass production in brassicaceous cover crop species is again minimal and concerns only a limited number of species. For *S. alba*, the proportion of roots to total biomass production was 28%, for *R. sativus* 38%, and for *Brassica rapa* 40%, respectively [3]. Liu et al. (2015) [2] evaluated aboveground and underground biomass production of brassicaceous cover crops (*R. sativus*, *S. alba,* and *Raphanus longipinnatus*) at various locations. The aboveground biomass production of these species was more than 4 times higher compared to the production of dry underground biomass. It is reported that the roots of *S. alba* constitute approximately 5–25% of the total plant biomass [40]. Thorup-Kristensen et al. (2001) [33] discovered that the ratio between dry aboveground biomass and underground biomass was 2.93 when *B. napus* (winter variety) was cultivated as a cover crop.

An important aspect to consider is evaluating the duration of intercrop vegetation, primarily influenced by the timing of sowing and the onset of intercrop maturity. It is indisputable that intercrop growth should be terminated as plants reach the initial stage of generative organ ripening to prevent the dispersal of mature intercrop seeds into the soil seed bank [21]. Nevertheless, these data remain unavailable in scientific literature. The primary objective of this study is to evaluate the production of both aboveground and underground biomass from non-traditional intercrop species within the *Brassicaceae* family, under Central European conditions, where the literature lacks information on their performance.

The following specific objectives were addressed within the scope of the study: (1) determine the influence of sowing date on above- and underground biomass of the evaluated species, (2) determine the vegetative to inflorescent biomass ratio and degradability, and (3) estimate the time to flowering for each evaluated species. The objectives of this study are based on the hypothesis that different *Brassicaceae* species respond differently to the sowing date in terms of biomass production and exhibit distinct stand development patterns in relation to the onset of growth stages. This information will help select brassicaceous intercrops for improved soil structure and carbon sequestration, as well as understand the best time to terminate these crops to gain their benefits.

## 2. Results

The results of average aboveground and underground biomass production (2021–2023) at the first evaluation date (BBCH 17–19) showed that at the first sowing date, the production of underground biomass ranged from 0.023 to 0.129 t/ha, and the dry aboveground biomass ranged from 0.266 to 1.006 t/ha (Table 1). *S. alba* and *R. sativus* demonstrated significantly higher underground biomass compared to above-ground production. Based on average values, the crops with the lowest production of underground and aboveground biomass included *C. sativa*, *C. abyssinica*, and *B. napus*. At the second sowing date, the production of underground biomass ranged from 0.079 to 0.631 t/ha, and the dry aboveground biomass ranged from 0.337 to 3.861 t/ha (Table 1). Based on statistical evaluation, no significant differences were proven between most of the average biomass production values. The biomass production evaluation at the third sowing date again indicated higher values of underground and aboveground biomass production for *S. alba* and *R. sativus*. Statistically significantly higher values of dry aboveground and underground biomass production in the BBCH 17–19 stages were observed for the crops *R. sativus* and *S. alba* (variety Paliisse), Table 1.

At the second evaluation date, for the aboveground biomass production (BBCH 63–67) corresponding to the end of vegetation before entering the generative organ formation phase, statistically significant differences between the average production of underground and aboveground biomass were demonstrated (Table 2). For the first sowing date, the varieties of *B. juncea* (VNIIMK 12), *S. alba* (Paliisse), and *R. sativus* (Lucas) had the highest values of aboveground and underground biomass; at the second sowing date, *S. alba* (Paliisse) and *R. sativus* (Lucas) again showed the highest values. For the third sowing date, the highest dry aboveground and underground biomass production was found in *C. abyssinica* and *R. sativus* (Lucas) (Table 2). For the fourth sowing date, the variety of *S. alba* (Paliisse) had the highest dry aboveground biomass. The average values of above-ground biomass production of *Brassicaceae* species at different sowing dates during the second evaluation period ranged from 0.932 to 11.853 t/ha.

Based on the methodology of Kwiatkowski et al. (2023) [20], where the authors calculated carbon fixation by the biomass of cruciferous cover crops using a carbon content value of 42%, it is also possible to determine the calculated carbon fixation of the evaluated stands (the data from Table 2 serve as the basis for the determination). According to the aforementioned calculation, the carbon content in the aboveground biomass of the monitored cover crop stands (average of the years for the given sowing date and sampling period BBCH 63–67) ranged from 1.47 to 5.95 t C per hectare for the first sowing date, from 1.27 to 4.41 t/ha for the second sowing date, from 0.47 to 4.63 t/ha for the third, and from 1.14 to 3.06 t C per hectare for the fourth.

Figure 1 shows a comparison of the dry aboveground biomass production of the evaluated species and their varieties in relation to the sowing date. The aboveground biomass production at the second to fourth sowing dates is compared to the aboveground biomass production achieved at the BBCH 63–67 stages at the first sowing date (100%). The results indicate that in most cases, the highest dry aboveground biomass production was achieved at the first sowing date.

Table 3 documents the average values of the ratio of dry aboveground to underground biomass at each evaluation date, depending on the sowing dates. At the first sowing date, the ratio of aboveground to underground biomass in the BBCH 17–19 stages ranged from 8.0 to 15.2. For the second sowing date, the ratio ranged from 3.7 to 9.2; for the third sowing date, from 8.2 to 13.8; and for the fourth sowing date, from 6.0 to 20.1. When evaluated at the BBCH 63–67 stage, the average ratio values ranged from 3.8 to 9.6 for the first sowing date, 4.6 to 10.1 for the second, 8.0 to 14.8 for the third, and 3.7 to 8.7 for the fourth (Table 3). On average, the share of root biomass production in the total biomass production for the evaluated species (average of years for the given sowing date and sampling period BBCH 63–67) reached 15.2% for the first sowing date, 5.5% for the second, 10.4% for the third, and 18.3% for the fourth.

Figure 2 shows the average values of the ratios of dry aboveground to underground biomass production at the BBCH 17–19 and BBCH 63–67 stages for all sowing dates over the period 2021–2023. The graph indicates that a wider ratio between aboveground and underground biomass is more common in the BBCH 17–19 stages.

Regression analyses confirmed a positive correlation between aboveground and underground biomass production in the BBCH 17–19 and BBCH 63–67 stages. Figure 3 illustrates the relationship between aboveground and underground biomass production in the BBCH 17–19 stages, incorporating the average biomass values of species from all four sowing dates over the 2021–2023 period. Figure 4 confirms a positive correlation between the production of aboveground and underground dry biomass for the evaluated *Brassicaceae* crops across the first to fourth sowing dates over the 2021–2023 period.

As part of the evaluation, the proportion of leaves in the total biomass production was monitored. Table 4 documents the average weight proportion of leaves in the total biomass production of crops depending on the sowing date for the period 2021–2023. The highest proportion of leaves in the aboveground biomass production was recorded for *B. napus* at the 2nd, 3rd, and 4th sowing dates. Plants of *B. napus* at the second sowing date rarely entered the elongation phase, while those sown at the third and fourth dates remained in the rosette leaf phase throughout the evaluation period. The highest proportion of leaves in the total aboveground biomass production was observed at the fourth sowing date, with plants predominantly entering the winter period in the rosette leaf phase or the elongation phase.

Effective vegetation length has also been assessed. In general, *Brassicaceae* cover crops required the longest time (days) from the sowing date to entering BBCH 69 at the first sowing date (Table 5). For *B. napus*, *C. abyssinica*, and *R. sativus*, this period was 100 days or more. The second sowing date was generally associated with a rapid transition of crops to BBCH 69, with the period from sowing to entering BBCH 69 ranging from 52 to 66 days. *C. abyssinica* crops required the longest time to enter BBCH 69. The third sowing date showed the minimum time needed to reach the BBCH 69 stage, which was 84 days. During the second and third sowing dates, *B. napus* plants did not enter the flowering phase due to the absence of the vernalization process. The fourth sowing date was not evaluated because no species entered BBCH 69.

## 3. Discussion

The average aboveground biomass values observed in the evaluated species are generally consistent with results reported in the literature, although available data are primarily limited to *S. alba* and *B. napus*. Information on the growth dynamics of non-traditional species under Central European conditions is lacking. Therefore, this study evaluates the timing of the plants’ transition to the generative phase. The reported values of above-ground biomass production for *S. alba* were within 0.5 to 4 t/ha [27]. Variations in leaf proportion within the plant canopy have also been described by Brant et al. (2022) [21], who evaluated these parameters across five varieties of *S. alba*. The results obtained in the present study are consistent with previously reported biomass values for *S. alba* (4.26 t/ha) and *B. napus* (3.40 t/ha) [28]. Across all sowing dates, fluctuations in above-ground biomass production ranged from 72.3% to 135.3% of the mean value. Data on the variation in cover crop yields depending on sowing time throughout the year are not available in the literature, but can be partly explained by weather-related differences between years [11,26].

The shoot-to-root ratio determined in this study ranged from 3.7 to 20.1 across the evaluated years, indicating considerable variability in underground biomass allocation among the tested species. Thorup-Kristensen et al. (2001) reported a shoot-to-root dry biomass ratio of 2.93 for *B. napus* [33], which corresponds to the lower end of the ratio observed here for more vigorous species such as *B. napus*, *R. sativus*, and *S. alba* when assessed at growth stages BBCH 63–67. In contrast, markedly higher ratios were found in stands sampled at earlier stages (BBCH 17–19), particularly in species with a smaller growth habit (e.g., *C. sativa* as shown in Table 3. The increase in shoot-to-root biomass ratio values at the BBCH 17–19 growth stages may have been influenced by the root washing method, as a considerable number of fine roots were present on the plants, which could not be quantitatively assessed using this technique.

At BBCH 63–67, the contribution of roots to the total biomass production ranged, depending on the sowing date, from 5.5 to 18.3%. These values are again lower than those reported in the available literature. Talgre et al. (2011) [3] reported that for selected cruciferous cover crop species, the share of underground biomass in total biomass production ranged from 28 to 40%, although this comparison involves taller, more robust species such as *S. alba*, *R. sativus*, and *B. rapa*. In the present study, root production was evaluated only to a depth of 0.2 m of the soil profile, which likely resulted in conservative estimates. The obtained results are consistent with the information reported by MESAM (2017) [40], stating that the roots of *S. alba* constitute approximately 5–25% of the total plant biomass. It is well established that shoot-to-root biomass ratio of crop stands is also significantly influenced by environmental and management factors [39]. The development of root systems is affected not only by soil conditions but also by the site’s moisture regime, which may impact root biomass production [41].

Although information on biomass partitioning among organs remains limited in the literature, these patterns are important in the context of soil conservation and cover cropping. However, understanding this parameter is relevant in the context of soil conservation technologies, as it can be assumed that leaf biomass contributes more substantially to soil cover during the vegetation period, thereby reducing erosion processes. The positive effect of soil cover on the mitigation of erosion is highlighted by Morgan (2005) [42]. Similarly, Brant et al. (2022) emphasize the beneficial role of plant leaves in providing soil cover but note that leaves decompose more rapidly after crop termination, with stems becoming the primary contributors to soil cover thereafter [21]. This accelerated degradation of leaves may be primarily related to their narrower C:N ratio compared to stems [22].

The observed species-specific differences suggest that the length of the growing season and the associated benefits of cover crops can be optimized through appropriate species selection. This approach allows for an extended period of positive effects, including soil protection and carbon input, as also noted by Brant et al. (2011) [11], Gaweda (2011), and Smit et al. (2019) [24,25].

## 4. Materials and Methods

Field experiments were conducted in Eastern Bohemia (49.9056481° N, 17.9100133° E, altitude 250 m above sea level) from 2021 to 2023. The soil texture at the given location can be characterized by a clay content of 5%, silt content of 71%, and sand content of 24%. The average organic matter content was 2.32%, and the pH was 6.7 for the period 2021 to 2023 (soil characteristics are based on regular measurements by OSEVA PRO Ltd., Opava, Czechia). Soil organic matter content was determined using the loss-on-ignition method. Soil pH was measured potentiometrically after preparing a soil-distilled water suspension at the standard ratio. Additional details are provided in the Appendix A. Average monthly precipitation totals and average air temperatures are documented in Figure 5 (source: www.chmi.cz, Opava station).

During the field experiments, nine species belonging to the *Brassicaceae* family were examined, with eight of them being evaluated with two varieties or de novo cultivars (refer to Table 6). These are the following species: *B. rapa* L. subsp. oleifera (DC.) Mentzger, *Brassica nigra* (L.) Koch, *Brassica juncea* (L.) Czern. et Cosson, *C. sativa* (L.) Crantz, *C. abyssinica,* and *E. sativa* (L.) Mill. in comparison with commonly cultivated species: *B. napus* L. subsp. *napus*, *S. alba* L., and *R. sativus* (Figure 6). *Brassicaceae* species were selected based on the confirmed ability to produce viable seeds at the study site. In terms of the complexity of the assessment, two different varieties or genetic sources for breeding were evaluated for most species at one location.

The sowing rates, represented by the number of seeds per unit area for each species, are detailed in Table 6, and these rates remained consistent across all sowing dates throughout the assessment years. The seeds used for the species originated from a single location, sourced from the seed production areas of OSEVA PRO Ltd. During each of the evaluated years, four sowing sessions were conducted (refer to Table 7). All sowing was done using a seeding machine with 125 mm row spacing. Winter cereals consistently served as the preceding crop for *Brassicaceae* cover crops, and primary soil tillage consisted of plowing. Soil preparation prior to sowing was completed on the same day as sowing. The crops were treated with insecticides to prevent pest damage, aiming to avoid plant reduction and leaf surface damage. No fertilization was applied before or during vegetation. The experiment was conducted in the form of trial plots measuring 12 × 1.5 m for each species and variety. Subsequently, the plots were divided into four pseudo replications, each measuring 1.5 × 3 m, which were evaluated individually, resulting in four replicates per species. The limitation in plot size was due to the restricted quantity of seed material, which originated primarily from the maintenance breeding of the gene pool. The average evaluation for the years 2021 to 2023 was calculated as the mean of the replicates across the individual years (i.e., 12 replicates in total). The selection of varietal materials was based on a prior screening conducted from 2018 to 2020 within the framework of maintenance breeding, which demonstrated the suitability of the selected varieties and species for sowing throughout the entire growing season. Within these plots, pseudo-replications were established to assess the number of plants per unit area and for plant sampling to determine aboveground and underground biomass production.

The evaluation dates for aboveground and underground biomass production for each sowing are documented in Table 7. The first and last sowing dates of the year represent the use of plants as cover crops for mulch production for spring and later-sown spring crops. The second and third sowing dates are intended for use as summer and post-harvest catch crops [21]. The number of plants was determined at the first assessment date in four replications within the experimental plot of each species along a 0.5-m row length at the center of the experimental plot. At each assessment location, three individual plants, including their root systems, were collected from the left side of the plot (relative to the center) during the first assessment date, resulting in a total of 12 plants per site. Plants were carefully excavated with the surrounding soil intact to a depth of 0.2 m to preserve root structure. After excavation, roots were gently washed to remove adhering soil without damaging fine root structures, ensuring accurate measurement of underground biomass. A root sampling depth of 0.2 m was chosen based on the topsoil profile, as sampling from deeper subsoil layers was technically challenging and risked significant root damage due to soil cracking. While this approach may underestimate total root biomass, it allows for a reliable assessment of root production within the topsoil layer.

Subsequently, the dry weight of both aboveground and underground biomass was determined for each plant (the biomass was dried at 105 °C for 48 h). At the second assessment date, again from the right side of each sampling location, three plants along with their roots were collected, totaling 12 plants from the species’ plot area. During the second sampling, the dry weight of the roots was determined, and for aboveground biomass, the dry weight of leaves and stems with inflorescences was determined. The assessment of aboveground and underground biomass production per unit area involved calculating both the average number of plants across the area and per individual plant. These calculated values were subsequently employed for statistical analysis. The ratio between underground and aboveground biomass was assessed for each plant sampled. Additionally, for each plant evaluated during the second assessment date, the proportion of leaf mass to the total aboveground biomass was determined. To manage the comprehensive datasets, average values of the monitored parameters for the period spanning 2021 to 2023 were incorporated into the study. During the growth of plant stands, BBCH stages were monitored at seven-day intervals. BBCH stages were subsequently determined to establish the effective length of vegetation. This length was determined from the sowing date to the date of plants reaching BBCH 69 (Biologische Bundesanstalt, Bundessortenamt und Chemische Industrie). Statistical analyses were performed using Statgraphic^®^ Plus software, version 4.0. Differences among treatments were assessed using analysis of variance (ANOVA), followed by Tukey’s post-hoc test to determine pairwise differences at a significance level of *p* < 0.05. Additionally, regression analyses were conducted using linear models to evaluate relationships between key variables, such as biomass production and sowing date. Data were checked for normality and homogeneity of variance before analysis to ensure the validity of the results.

## 5. Conclusions

In conclusion, this study demonstrates that both traditional and non-traditional *Brassicaceae* species can produce significant aboveground and underground biomass under Central European conditions, with *S. alba* and *R. sativus* showing the highest biomass yields. Biomass allocation varied among species and growth stages, with shoot-to-root ratios ranging widely and fine root recovery influencing measured values. Species-specific differences in leaf and stem biomass highlight their distinct contributions to soil cover and potential for erosion mitigation, while root biomass plays a key role in carbon sequestration. Sowing date strongly affected biomass production and phenological development, underscoring the importance of optimizing planting time to maximize the ecological and agronomic benefits of these cover crops. Overall, the findings indicate that careful selection of *Brassicaceae* species and sowing strategies can enhance sustainable cropping systems by improving soil protection, nutrient retention and organic carbon inputs.

## Figures and Tables

**Figure 1 plants-14-03654-f001:**
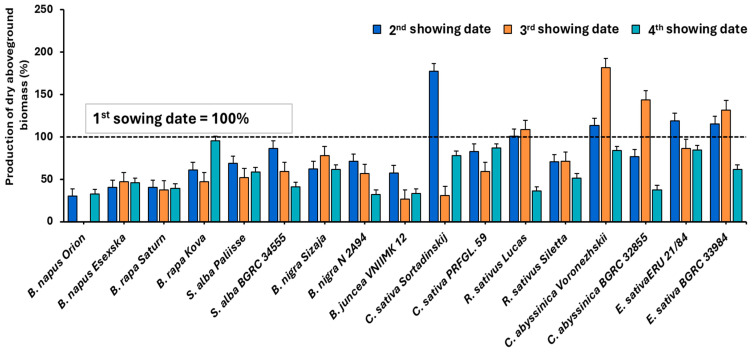
Comparison of dry aboveground biomass production between sowing dates. Biomass from the first sowing date is set as 100%. (BBCH 63–67, mean values for 2021–2023), vertical bars represent standard errors.

**Figure 2 plants-14-03654-f002:**
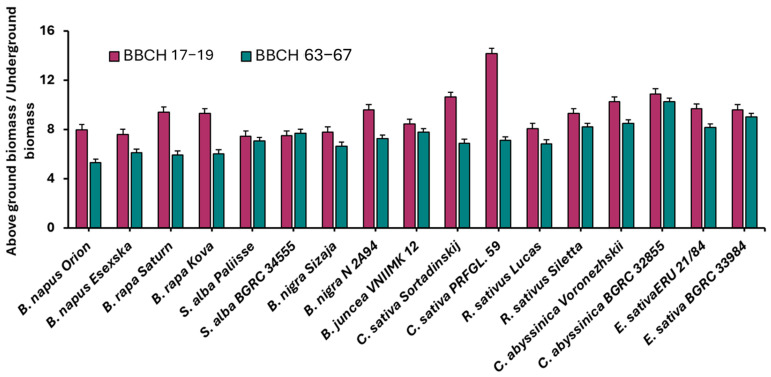
Average ratio of dry aboveground to underground biomass (AgB/UgB) at growth stages BBCH 17–19 and BBCH 63–67 for the period 2021–2023, vertical bars represent standard errors.

**Figure 3 plants-14-03654-f003:**
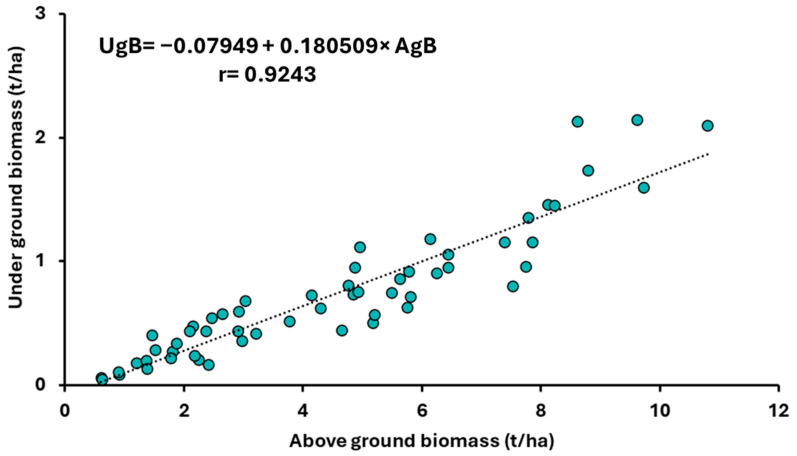
Relationship between aboveground (AgB, t ha^−1^) and underground (UgB, t ha^−1^) biomass at growth stage BBCH 17–19, based on average values of both components across all four sowing dates during the period 2021–2023.

**Figure 4 plants-14-03654-f004:**
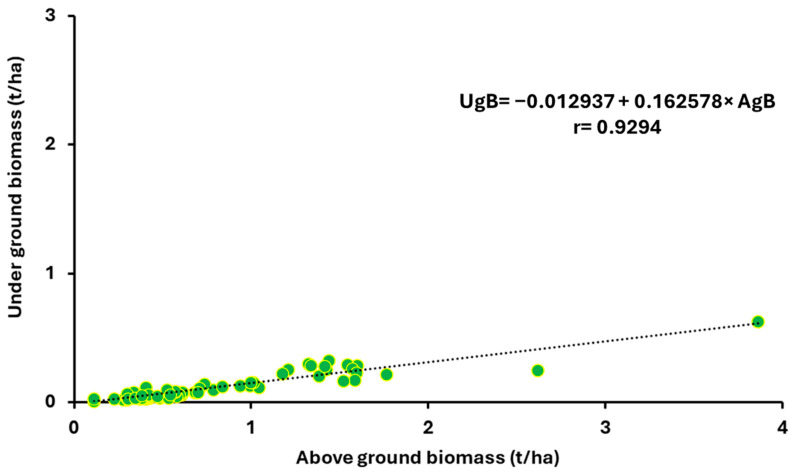
Relationship between aboveground (AgB, t ha^−1^) and underground (UgB, t ha^−1^) biomass at growth stages BBCH 63–67, based on average values of both components across all four sowing dates during the period 2021–2023.

**Figure 5 plants-14-03654-f005:**
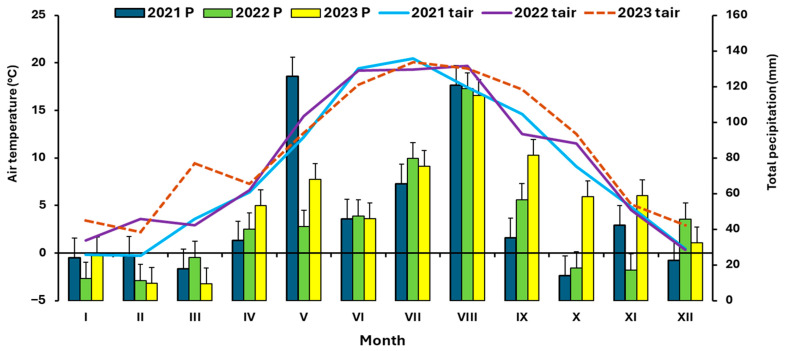
Monthly mean air temperature and total precipitation from 2021 to 2023.

**Figure 6 plants-14-03654-f006:**
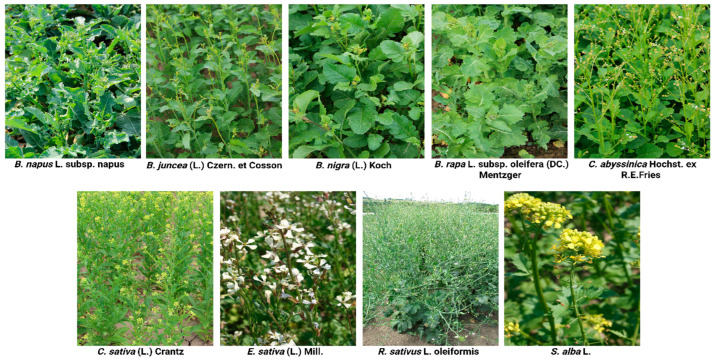
Representative images of catch crop species evaluated in the study.

**Table 1 plants-14-03654-t001:** Production of dry underground biomass (UgB, t/ha) and aboveground biomass (AgB, t/ha) of selected species from the *Brassicaceae* family depending on sowing time, averaged for the years 2021–2023. The assessment was carried out at the BBCH stages 17–19. Different indices among the averages indicate a statistically significant difference at the 95% significance level (ANOVA, Tukey).

		1. Sowing Date				2. Sowing Date				3. Sowing Date				4. Sowing Date			
Species	Variety/New Breeding	UgB (t/ha)		AgB (t/ha)		UgB (t/ha)		AgB (t/ha)		UgB (t/ha)		AgB (t/ha)		UgB (t/ha)		AgB (t/ha)	
*B. napus*	Orion	0.044	abc	0.374	ab	0.079	a	0.337	a	0.040	ab	0.289	abcd	0.057	ab	0.498	ab
*B. napus*	Esexska	0.055	abcd	0.503	abc	0.118	a	0.405	a	0.025	a	0.211	ab	0.078	abc	0.568	ab
*B. rapa*	Saturn	0.058	abcd	0.573	bc	0.258	a	1.208	ab	0.030	a	0.265	abc	0.084	abc	0.571	ab
*B. rapa*	Kova	0.047	abc	0.445	ab	0.290	a	1.595	ab	0.038	a	0.278	abcd	0.084	abc	0.707	ab
*S. alba*	Paliisse	0.129	f	0.977	d	0.294	a	1.544	ab	0.064	ab	0.506	bcde	0.131	bc	0.865	b
*S. alba*	BGRC 34555	0.110	ef	1.006	d	0.329	a	1.437	ab	0.078	ab	0.591	de	0.144	c	0.733	ab
*B. nigra*	Sizaja	0.077	cde	0.591	bc	0.301	a	1.322	ab	0.067	ab	0.270	abcd	0.096	abc	0.522	ab
*B. nigra*	N 2A94	0.057	abcd	0.596	bc	0.256	a	1.424	ab	0.028	a	0.204	ab	0.059	abc	0.419	a
*B. juncea*	VNIIMK 12	0.076	bcde	0.602	bc	0.228	a	1.177	ab	0.039	ab	0.345	abcd	0.109	abc	0.707	ab
*C. sativa*	Sortadinskij	0.031	a	0.386	ab	0.266	a	1.570	ab	0.013	a	0.097	a	0.045	ab	0.406	a
*C. sativa*	PRFGL. 59	0.035	ab	0.426	ab	0.154	a	1.012	ab	0.041	ab	0.106	a	0.039	a	0.613	ab
*R. sativus*	Lucas	0.129	f	0.924	d	0.631	b	3.861	c	0.135	b	1.122	f	0.284	d	1.412	c
*R. sativus*	Siletta	0.097	def	0.756	cd	0.230	a	1.591	ab	0.137	b	0.717	e	0.290	d	1.334	c
*C. abyssinica*	Voronezhskii	0.023	a	0.266	a	0.253	a	2.616	bc	0.060	ab	0.547	cde	0.055	ab	0.383	a
*C. abyssinica*	BGRC 32855	0.041	abc	0.419	ab	0.176	a	1.583	ab	0.061	ab	0.510	bcde	0.062	abc	0.540	ab
*E. sativa*	ERU 21/84	0.044	abc	0.400	ab	0.221	a	1.763	ab	0.041	ab	0.421	abcde	0.048	ab	0.470	ab
*E. sativa*	BGRC 33984	0.030	a	0.366	ab	0.207	a	1.381	ab	0.030	a	0.296	abcd	0.079	abc	0.699	ab
	*p*-Value	0.0000		0.0000		0.0000		0.0000		0.0000		0.0000		0.0000		0.0000	

**Table 2 plants-14-03654-t002:** Production of dry underground biomass (UgB, t/ha) and aboveground biomass (AgB, t/ha) of selected species from the *Brassicaceae* family depending on sowing time, average for the years 2021–2023. The assessment was carried out at the BBCH stages 63–67. Different indices among the averages indicate a statistically significant difference at the 95% significance level (ANOVA, Tukey).

		1. Sowing Date				2. Sowing Date				3. Sowing Date				4. Sowing Date			
Species	Variety/New Breeding	UgB (t/ha)		AgB (t/ha)		UgB (t/ha)		AgB (t/ha)		UgB (t/ha)		AgB (t/ha)		UgB (t/ha)		AgB (t/ha)	
*B. napus*	Orion	1.956	de	8.129	bcde	0.595	ab	2.431	a	*		*		0.424	a	2.648	a
*B. napus*	Esexska	1.992	de	7.559	abcde	0.934	abc	3.044	abc	0.548	abc	3.552	a	0.863	ab	3.473	ab
*B. rapa*	Saturn	1.257	abcd	6.458	abcd	0.574	ab	2.594	ab	0.265	a	2.407	a	0.653	a	2.551	a
*B. rapa*	Kova	0.977	abc	4.834	abc	0.626	ab	2.957	abc	0.331	ab	2.261	a	0.922	ab	4.601	ab
*S. alba*	Paliisse	1.692	cde	10.308	de	1.148	bc	7.080	cd	0.587	abc	5.330	abc	1.251	abc	6.042	b
*S. alba*	BGRC 34555	1.249	abcd	7.926	bcde	1.023	abc	6.857	bcd	0.451	ab	4.669	ab	0.685	a	3.255	ab
*B. nigra*	Sizaja	1.161	abcd	7.799	bcde	0.819	ab	4.865	abcd	0.722	abcd	6.054	abc	1.097	ab	4.795	ab
*B. nigra*	N 2A94	1.486	bcde	8.535	bcde	0.918	ab	6.061	abcd	0.483	abc	4.841	abc	0.483	a	2.731	a
*B. juncea*	VNIIMK 12	2.310	e	11.853	e	0.989	abc	6.830	bcd	0.304	ab	3.136	a	1.010	ab	3.961	ab
*C. sativa*	Sortadinskij	0.463	a	3.040	a	0.813	ab	5.390	abcd	0.184	a	0.932	a	0.357	a	2.364	a
*C. sativa*	PRFGL. 59	0.745	ab	4.133	abc	0.438	a	3.427	abc	0.278	ab	2.436	a	1.060	ab	3.581	ab
*R. sativus*	Lucas	1.751	cde	8.866	cde	1.562	c	8.928	d	1.246	d	9.614	bc	2.100	c	3.189	ab
*R. sativus*	Siletta	1.510	bcde	8.253	bcde	0.782	ab	5.805	abcd	0.562	abc	5.859	abc	1.685	bc	4.248	ab
*C. abyssinica*	Voronezhskii	0.898	abc	5.288	abc	0.651	ab	5.988	abcd	1.063	cd	9.601	bc	0.535	a	4.420	ab
*C. abyssinica*	BGRC 32855	0.911	abc	7.026	abcd	0.526	a	5.373	abcd	0.940	bcd	10.089	c	0.358	a	2.647	a
*E. sativa*	ERU 21/84	0.783	ab	5.056	abc	0.747	ab	6.015	abcd	0.494	abc	4.361	a	0.565	a	4.270	ab
*E. sativa*	BGRC 33984	0.665	ab	4.510	abc	0.583	ab	5.196	abcd	0.454	ab	5.936	abc	0.454	a	2.783	a
	*p*-Value	0.0000		0.0000		0.0000		0.0000		0.0000		0.0000		0.0000		0.0001	

* The evaluation was not conducted in 2022 due to pest damage.

**Table 3 plants-14-03654-t003:** Average values of the ratio between aboveground dry biomass (AgB, t/ha) and underground dry biomass (UgB, t/ha) of cruciferous crops in relation to sowing time and BBCH growth stage. Averaged for the years 2021–2023. Different indices between averages indicate statistically significant differences between the means at a 95% significance level (ANOVA, Tukey).

								Ratio of Aboveground and Underground Biomass (AgB/UgB)									
Species	Variety/New Breeding	BBCH 17–19								BBCH 63–67							
		1. Sowing Date		2. Sowing Date		3. Sowing Date		4. Sowing Date		1. Sowing Date		2. Sowing Date		3. Sowing Date		4. Sowing Date	
*B. napus*	Orion	9.5	a	4.8	ab	8.7	ab	8.9	abcd	4.3	ab	5.6	abc			6.0	abcde
*B. napus*	Esexska	10.3	ab	3.7	a	8.2	a	8.1	abc	3.8	a	5.7	abc	10.8	ab	4.1	ab
*B. rapa*	Saturn	11.6	ab	4.6	ab	12.9	ab	8.5	abcd	5.5	abc	4.6	a	9.3	ab	4.4	ab
*B. rapa*	Kova	10.3	ab	6.3	abcd	10.1	ab	10.3	bcd	5.5	abc	5.3	ab	8.4	a	5.0	abc
*S. alba*	Paliisse	8.8	a	5.2	abc	8.5	a	7.3	abc	6.2	bc	6.1	abc	10.8	ab	5.1	abc
*S. alba*	BGRC 34555	10.1	ab	5.6	abc	8.3	a	5.8	a	7.1	cd	6.6	abcd	11.5	ab	5.6	abcd
*B. nigra*	Sizaja	8.5	a	5.1	ab	8.8	ab	8.7	abcd	7.2	cd	6.4	abc	8.1	a	4.8	abc
*B. nigra*	N 2A94	15.0	b	5.6	abc	8.9	ab	8.9	abcd	6.1	abc	6.8	abcd	10.1	ab	5.8	abcde
*B. juncea*	VNIIMK 12	8.9	a	5.2	abc	10.2	ab	9.4	abcd	6.0	abc	6.9	bcd	14.0	ab	4.2	ab
*C. sativa*	Sortadinskij	15.2	b	7.2	bcde	8.7	ab	11.3	cd	6.7	cd	6.0	abc	8.0	a	6.7	bcde
*C. sativa*	PRFGL. 59	14.9	b	7.9	cde	13.8	ab	20.1	e	5.9	abc	7.0	bcd	9.5	ab	6.0	abcde
*R. sativus*	Lucas	8.0	a	6.0	abc	12.4	ab	6.0	a	5.3	abc	5.8	abc	12.6	ab	3.7	a
*R. sativus*	Siletta	8.6	a	7.1	bcde	15.2	b	6.2	ab	5.7	abc	7.3	bcd	14.8	b	5.0	abc
*C. abyssinica*	Voronezhskii	11.5	ab	9.2	d	11.0	ab	9.3	abcd	6.8	cd	9.7	e	9.1	ab	8.2	de
*C. abyssinica*	BGRC 32855	11.0	ab	8.8	cde	11.3	ab	12.4	d	9.0	d	10.1	e	13.3	ab	8.7	e
*E. sativa*	ERU 21/84	10.2	ab	7.0	bcde	10.8	ab	10.6	cd	6.8	cd	7.9	cde	10.3	ab	7.6	cde
*E. sativa*	BGRC 33984	12.9	ab	6.1	abc	10.8	ab	8.6	abcd	7.5	cd	8.9	de	13.7	ab	6.0	abcde
	*p*-Value	0.0000		0.0000		0.0000		0.0000		0.0000		0.0000		0.0041		0.000	

**Table 4 plants-14-03654-t004:** The percentage of leaf biomass (%) in the total aboveground biomass of the crop (with the total crop weight being the sum of the dry biomass of leaves, stems, and inflorescences) during the BBCH 63–67 growth stages (averaged over the years 2021–2023).

Species	Variety/New Breeding	1. Sowing Date	2. Sowing Date	3. Sowing Date	4. Sowing Date
		Percentage of Leaf Biomass (%) in the Total Aboveground Biomass of the Crop			
*B. napus*	Orion	41.6	81.8	100.0	100.0
*B. napus*	Esexska	46.2	84.7	94.9	100.0
*B. rapa*	Saturn	24.4	20.7	51.8	68.9
*B. rapa*	Kova	33.8	22.5	55.0	88.2
*S. alba*	Paliisse	30.1	30.7	51.9	53.3
*S. alba*	BGRC 34555	23.4	28.8	45.2	64.5
*B. nigra*	Sizaja	29.0	29.4	42.1	75.0
*B. nigra*	N 2A94	32.1	36.0	46.4	81.0
*B. juncea*	VNIIMK 12	31.5	28.9	55.5	86.6
*C. sativa*	Sortadinskij	36.0	38.1	47.1	67.9
*C. sativa*	PRFGL. 59	41.3	47.1	38.4	85.8
*R. sativus*	Lucas	44.8	44.5	52.2	100.0
*R. sativus*	Siletta	45.2	40.9	26.8	98.1
*C. abyssinica*	Voronezhskii	43.5	59.0	60.3	90.4
*C. abyssinica*	BGRC 32855	43.0	62.5	60.3	89.1
*E. sativa*	ERU 21/84	37.6	50.0	76.2	83.6
*E. sativa*	BGRC 33984	37.1	38.3	67.9	78.5

**Table 5 plants-14-03654-t005:** The number of days from the sowing date until the crops reach the BBCH 69 growth stage (averaged for the years 2021–2023). The evaluation covers the first to third sowing dates. For the fourth sowing date, the plants reached a maximum of BBCH 63.

Species	Variety/New Breeding	1. Sowing Date	2. Sowing Date	3. Sowing Date
*B. napus*	Orion	113	*****	*****
*B. napus*	Esexska	113	*****	*****
*B. rapa*	Saturn	91	52	84
*B. rapa*	Kova	91	52	84
*S. alba*	Paliisse	94	54	87
*S. alba*	BGRC 34555	94	54	87
*B. nigra*	Sizaja	94	54	85
*B. nigra*	N 2A94	94	54	85
*B. juncea*	VNIIMK 12	94	54	85
*C. sativa*	Sortadinskij	94	56	84
*C. sativa*	PRFGL. 59	91	56	84
*R. sativus*	Lucas	100	66	84
*R. sativus*	Siletta	100	58	84
*C. abyssinica*	Voronezhskii	100	62	84
*C. abyssinica*	BGRC 32855	100	62	84
*E. sativa*	ERU 21/84	95	59	87
*E. sativa*	BGRC 33984	95	59	87

**Table 6 plants-14-03654-t006:** The varieties and de novo cultivars, along with their respective sowing rates (seeds per m^2^) of each species used in the study.

Species	Variety/New Breeding *	Sowing Rate (Seeds pre m^2^)
*B. napus* L. subsp. *napus*	Orion, Esexska	89
*B. rapa* L. subsp. *oleifera* (DC.) Mentzger	Saturn, Kova	178
*S. alba* L.	Paliisse, BGRC 34555 *	89
*B. nigra* (L.) Koch	Sizaja, N 2A94 *	148
*B. juncea* (L.) Czern. et Cosson	VNIIMK 12 *	148
*C. sativa* (L.) Crantz	Sortadinskij, PRFGL. 59 *	178
*R. sativus* L. *oleiformis*	Lucas, Siletta	89
*C. abyssinica* Hochst. ex R.E.Fries	Voronezhskii, BGRC 32855 *	148
*E. sativa* (L.) Mill.	ERU 21/84 *, BGRC 33984 *	178

* new breeding.

**Table 7 plants-14-03654-t007:** Sowing dates of crops and dates for assessing biomass production and time (days after the sowing) over the years 2021–2023.

No.		Date in 2021			2022			2023	
	Sowing Date	Biomass Assessment Date		Sowing Date	Biomass Assessment Date		Sowing Date	Biomass Assessment Date	
		BBCH * 17–19/No of Days After Sowing	BBCH 63–67/No of Days After Sowing		BBCH 17–19/No of Days After Sowing	BBCH 63–67/No of Days After Sowing		BBCH 17–19/No of Days After Sowing	BBCH 63–67/No of Days After Sowing
**1.**	29.3.	20.5./52	4.6./67	23.9.	10.5./48	6.6./78	27.3.	9.5./43	29.5./63
**2.**	26.5.	28.6./31	8.7./43	18.5.	13.6./29	27.6./43	11.5.	12.6./32	4.7./54
**3.**	28.6.	27.7./31	2.9./68	27.6.	20.7./23	30.8./64	4.7.	20.7./16	10.8./37
**4.**	6.9.	4.10./25	9.11/59	3.9.	17.10./44	8.11./66	5.9.	17.10./42	8.11./64

* The range of BBCH stages was determined due to the varying onset of crop development into a specific growth period.

## Data Availability

The data that support the findings of this study are available from the first author, [V.B.], upon reasonable request.

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
