# Peer review of "The Effect of Sowing Date on the Biomass Production of Non-Traditional and Commonly Used Intercrops from the Brassicaceae Family"

_plants, 2025, doi:10.3390/plants14233654_

Round 1
Reviewer 1 Report
Comments and Suggestions for Authors
-provide a combined picture showing the photos of various catch crops in this study.
-the figures in this study do not meet the publication standards.
-place the figures and tables in the body of the text instead of at the end of the results.
Author Response
Authors thank the reviewers for their careful reading and constructive comments. We are pleased to confirm that the suggestions have been accepted and incorporated into the revised manuscript.
Reviewer 1:
Comment 1: Provide a combined picture showing the photos of various catch crops in this study.
Response: We appreciated this helpful suggestion. We have prepared a combined figure that displays representative photographs of all catch crops included in the study. This new figure has been added as Figure 2 in the revised version.
Comment 2: The figures in this study do not meet the publication standards.
Response: Thank you for pointing this out. All figures have been revised to improve resolution, labelling and overall clarity to meet the journal’s publication standards. We believe the improved versions enhance the readability and presentation of the results.
Comment 3: Place the figures and tables in the body of the text instead of at the end of the results.
Response: We agree with this recommendation. All figures and tables have now been relocated to appear within the main text at appropriate locations corresponding to their first citation. We believe that this change improves the flow and readability of the manuscript.
Reviewer 2 Report
Comments and Suggestions for Authors
The manuscript presents a substantial and valuable dataset on the biomass production of nine Brassicaceae species. However, the manuscript's writing and overall presentation have several shortcomings that need to be addressed. Detailed comments and specific suggestions for revision are provided in the annotated manuscript.

Author Response
Reviewer 2:
Comments and Suggestions for Authors The manuscript presents a substantial and valuable dataset on the biomass production of nine Brassicaceae species. However, the manuscript's writing and overall presentation have several shortcomings that need to be addressed. Detailed comments and specific suggestions for revision are provided in the annotated manuscript.
We sincerely thank the reviewer for the valuable comments and suggestions. We have carefully revised the manuscript accordingly. Below, we provide detailed responses to each point raised.
Comment 1: Abstract: The Conclusion section needs to be rewritten. It should focus on highlighting the key innovations of the study and providing a clear future outlook.
Response: The Conclusion section for the Abstract has been revised as recommended. It now emphasizes the key findings and their implications as follows: Overall, the results demonstrate that sowing date is a critical factor influencing growth dynamics and reproductive development in these underutilized Brassicaceae species. By identifying optimal planting windows, this study contributes to improved management strategies aimed at maximizing biomass production while supporting sustainable cropping practices, including enhanced soil organic carbon stabilization and reduced nutrient losses.
Comment 2: Introduction: The Introduction section requires revision for better structure and flow. Please organize it into 4-5 paragraphs as follows: Paragraphs 1-3/4: Establish the research background and discuss the relevant progress and current state of the field. Final Paragraph: Clearly state the research objectives and the specific scientific questions this study aims to address.
Response: This comment was also highlighted by another reviewer, and the suggested restructuring has been fully implemented. The Introduction is now reorganized into several clearly defined paragraphs, culminating with a concise statement of the study’s objectives.
Comment 3: The Materials and Methods section should also be broken down into separate subsections with clear subheadings to describe the experimental procedures and the statistical analysis, respectively.
Response: The materials and methods section has been reorganized into clearly separated subsections, including a distinct statistical analysis section. Tables and figures have been strategically inserted to improve structure and readability.
Comment 4: The Results section should focus exclusively on presenting your findings. Please move any citations to the Discussion section, where your results are interpreted and compared with existing literature.
Response: All citations and interpretative text have been removed from the Results section and relocated to the Discussion, which now provides comparative context and interpretation of the findings.
Comment 5: The figures and tables should not be grouped together at the end of the Result. According to the submission guidelines, please insert each figure and table into the text immediately after it is first mentioned in the corresponding results section.
Response: This formatting issue has been corrected in accordance with the submission guidelines. Each figure and table now appears directly after its initial reference in the Results text.
Comment 6: The Discussion section requires a major revision. It is currently underdeveloped and lacks a clear logical flow. We recommend rewriting this section to build a more coherent and compelling argument based on your findings.
Response: The discussion section has been extensively rewritten. We believe, it now provides a more coherent narrative, integrates relevant literature more effectively, and more clearly articulates the implications of the results.
Reviewer 3 Report
Comments and Suggestions for Authors
- Abstract: The innovation of this study should be clearly highlighted, such as stating that it is "the first systematic evaluation of biomass characteristics for six non-traditional Brassicaceae species in Central Europe." Key results should be quantified (e.g., providing specific biomass ranges for Sinapis alba and Raphanus sativus) to improve readability.
- Keywords: It is recommended to simplify "underground biomass vs. above-ground biomass ratio" to "root-to-shoot ratio" to avoid redundancy. Additional keywords such as "non-traditional Brassicaceae" or "sowing date optimization" could be included to better emphasize the research focus.
- Introduction: The existing research foundation for non-traditional species should be supplemented. For example, while the drought tolerance of Crambe abyssinica has been mentioned by Samarappuli et al. (2020), its performance in Central European conditions has not been reported. Similarly, the biomass potential of Eruca sativa in Southern European studies can be contrasted with the conditions of this study in Central Europe to highlight the necessity of the research.
- Introduction: The selection criteria for "non-traditional species" (e.g., ecological adaptability, gaps in previous research) should be clearly explained to avoid any perception of arbitrary species selection.
- Root Sampling Depth: The rationale for sampling roots to a depth of 0.2 meters (e.g., equipment limitations or species-specific root distribution characteristics) should be explicitly stated. It should also be acknowledged that this may underestimate deep root biomass, setting the stage for discussing methodological limitations.
- Discussion: For species with low biomass (e.g., C. abyssinica, B. napus), explanations should incorporate their biological traits (e.g., long growth cycles, late maturity) or environmental adaptability (e.g., sensitivity to Central European climate), rather than merely describing the phenomena.
- Language and Terminology: Some sentences are overly long and logically loose, and there is inconsistency in terminology (e.g., "underground" vs. "belowground"). These issues should be addressed to improve clarity and coherence.
The topic of this paper is valuable, but it requires refinement in experimental method details, correction of data presentation (e.g., table formatting, statistical parameters), deeper interpretation of results (e.g., species-specific responses, methodological limitations), and the addition of practical application recommendations. After revision, the paper can more fully demonstrate the potential of non-traditional Brassicaceae species in sustainable agriculture.
Author Response
Comment 1:
Abstract: The innovation of this study should be clearly highlighted, such as stating that it is "the first systematic evaluation of biomass characteristics for six non-traditional Brassicaceae species in Central Europe." Key results should be quantified (e.g., providing specific biomass ranges for Sinapis alba and Raphanus sativus) to improve readability.
Response: We appreciate your comments, and the abstract has been revised accordingly:
L 13: This study presents the first systematic evaluation of biomass characteristics for six non-traditional Brassicaceae species under Central European conditions, alongside commonly cultivated representatives of the family.
L 21: The results revealed significant species-specific differences in biomass accumulation.
L25: A positive correlation between aboveground and underground biomass was observed across species, though root-to-shoot ratios varied, influencing carbon allocation patterns and soil organic matter inputs.
L 30: The findings provide new insights into the biomass potential of underutilized Brassicaceae species in sustainable cropping systems, …
Comment 2:
Keywords: It is recommended to simplify "underground biomass vs. above-ground biomass ratio" to "root-to-shoot ratio" to avoid redundancy. Additional keywords such as "non-traditional Brassicaceae" or "sowing date optimization" could be included to better emphasize the research focus.
Response: Thank you for the suggestion, we have changed the keywords accordingly: aboveground biomass production; carbon fixation; leaf–stem ratio; non-traditional Brassicaceae; root-to-shoot ratio; sowing date optimization
Comment 3: Introduction: The existing research foundation for non-traditional species should be supplemented. For example, while the drought tolerance of Crambe abyssinica has been mentioned by Samarappuli et al. (2020), its performance in Central European conditions has not been reported. Similarly, the biomass potential of Eruca sativa in Southern European studies can be contrasted with the conditions of this study in Central Europe to highlight the necessity of the research.
Response: Thank you for your comment, we have changed the text accordingly:
Although drought and soil salinity tolerance of C. abyssinica has been documented in study of Samarappuli et al. (2020) [17], its performance under Central European conditions remains unreported. Similarly, Eruca sativa has shown promising biomass potential in South Europe, but data under central European climatic and soil conditions are lacking. These gaps highlight the need for systematic evaluation of non-traditional Brassicaceae species in this region.
Comment 4: Introduction: The selection criteria for "non-traditional species" (e.g., ecological adaptability, gaps in previous research) should be clearly explained to avoid any perception of arbitrary species selection.
Response: Thank you for raising this question.
Brassicaceae species were selected based on the confirmed ability to produce viable seeds at the study site. Information on biomass production is lacking for these species in their native regions, and they are not currently used as intercrops (catch crops).
Comment 5: Root Sampling Depth: The rationale for sampling roots to a depth of 0.2 meters (e.g., equipment limitations or species-specific root distribution characteristics) should be explicitly stated. It should also be acknowledged that this may underestimate deep root biomass, setting the stage for discussing methodological limitations.
Response: Yes, we completely agree with you. Therefore, we explained this sampling in a deeper detail:
A root sampling depth of 0.2 m was chosen based on the topsoil profile, as sampling from deeper subsoil layers was technically challenging and risked significant root damage due to soil cracking. While this approach may underestimate total root biomass, it allows for reliable assessment of root production within the topsoil layer.
Comment 6: Discussion: For species with low biomass (e.g., C. abyssinica, B. napus), explanations should incorporate their biological traits (e.g., long growth cycles, late maturity) or environmental adaptability (e.g., sensitivity to Central European climate), rather than merely describing the phenomena.
Response:
We sincerely thank the reviwer for this comment. Information on the growth dynamics of non-traditional species under Central European conditions is lacking. Therefore, this study evaluates the timing of the plants’ transition to the generative phase. For B. napus, the observed timing reflects the absence of vernalization, as this species is winter-hardy.
Comment 7: Language and Terminology: Some sentences are overly long and logically loose, and there is inconsistency in terminology (e.g., "underground" vs. "belowground"). These issues should be addressed to improve clarity and coherence.
Response: We thank the reviewer for this observation. The manuscript has been carefully revised to improve sentence structure and clarity, and terminology has been standardized by unifying the use of „belowground“ throughout the text.
Comment 8: The topic of this paper is valuable, but it requires refinement in experimental method details, correction of data presentation (e.g., table formatting, statistical parameters), deeper interpretation of results (e.g., species-specific responses, methodological limitations), and the addition of practical application recommendations. After revision, the paper can more fully demonstrate the potential of non-traditional Brassicaceae species in sustainable agriculture.
Response: We sincerely thank the reviewer for the constructive feedback. We have carefully revised the manuscript to provide more detailed descriptions of the experimental methods, improved data presentation (mainly figures and tables), expanded the interpretation of results and highlighted species-specific response. We believe these revisions enhance the clarity and applicability of the study.
Reviewer 4 Report
Comments and Suggestions for Authors
Commennts for Authors
The manuscript presents a well-designed, multi-year field study that provides valuable insights into biomass production dynamics of Brassicaceae intercrops under Central European conditions. The research question is clear and relevant, and the dataset collected over three seasons contributes to understanding both above- and below-ground biomass relationships in cover crops.
However, to enhance clarity, several revisions are recommended.
I kindly ask the Authors to read the suggestions below.
General note
- The last keyword (“underground biomass vs. aboveground biomass ratio “) repeats the first two. Please consider replacing it with another one that appropriately fits the content of the manuscript.
- Please pay attention to the correctness of Latin plant names and the abbreviation of duplicate species names. Please ensure that all Latin names are italicized.
- Chapters 1 and 2 are written as a single text string, making them difficult to read. Please divide the text into paragraphs.
- Please pay attention to the appropriate form of citations in the text and correct them.
- Why is more than half of the cited literature older than 10 years, and many of them date back even further? Please update the References.
Materials and Methods
- The list of tested plants (line 124-127) should be included in the Material and Methods section.
- There is a lack of information on the content of macro and micronutrients in soil. Enriching soil with these nutrients through intercrops is not the subject of research, but in the reviewer's opinion, a brief description and analysis of these nutrients would be desirable for agricultural research.
- The sampling method is unclear. Please describe it in more detail.
- There is no need to constantly repeat information about the years in which the experiments were conducted.
- Please specify what stage of plant development is indicated by the given BBCH scale grades (17-19 and 63-67).
- Figures and tables should appear as close as possible to the point where they are first mentioned in the text. Please place Tables 1 and 2 in their appropriate places in Section 2.
- Statistical methods used and the research results definitely require more detailed description .
Results
- Please place tables and figures in the appropriate place in the text. Please ensure that all tables and figures are referenced in the text.
- In the reviewer's opinion, it is also worthwhile to present the results regarding carbon fixation graphically.
- The results of the statistical analyses also require more detailed description and should be appropriately summarized/analyzed.
- Statements such as: "Differences in the proportion of leaves on the plant are also described by Brant et al (2022) [21], (...)" - lines 275, 276, or: "The mentioned differences (...), as pointed out by Brant et al. (...)" - lines 286-289, constitute a discussion and should be moved to chapter 4.
- Tables 3, 4, and 5 contain a lot of information (both biomass yield results and statistical analysis results) and are therefore quite difficult to read. Please consider presenting the results and analyses more clearly.
Discussion
- Please see note 4 in the Results section or consider combining the Results and Discussion sections into one coherent whole.
Conclusions
There is no section devoted to results in the manuscript! The summary should present substantive and synthetic conclusions based on the obtained research results, as well as discussions, if any.
Author Response
Averall comment:
The manuscript presents a well-designed, multi-year field study that provides valuable insights into biomass production dynamics of Brassicaceae intercrops under Central European conditions. The research question is clear and relevant, and the dataset collected over three seasons contributes to understanding both above- and below-ground biomass relationships in cover crops.
However, to enhance clarity, several revisions are recommended.
I kindly ask the Authors to read the suggestions below.
Response: We sincerely thank the reviewer for overall positive evaluation of our manuscript and for acknowledging the relevance of our Research and the value of the multi-year dataset. We have carefully considered all the suggested revisions and have accepted most for the reviewers’ comments, implementing changes to improve clarity, data presentation, and discussion of the results throughout the manuscript. We greatly appreciate the reviewers constructive feedback, which has helped enhance the quality and readability of our work.
Comment 1: The last keyword (“underground biomass vs. aboveground biomass ratio “) repeats the first two. Please consider replacing it with another one that appropriately fits the content of the manuscript.
Response: Thank you for this comment. Keywords have been revised to “ aboveground biomass production; carbon fixation; leaf–stem ratio; non-traditional Brassicaceae; root-to-shoot ratio; sowing date optimization“
Comment 2: Please pay attention to the correctness of Latin plant names and the abbreviation of duplicate species names. Please ensure that all Latin names are italicized.
Response: We thank the reviewer for this observation and have revised the text accordingly. All Latin names have been checked, corrected and consistently italicized throughout the manuscript.
Comment 3: Chapters 1 and 2 are written as a single text string, making them difficult to read. Please divide the text into paragraphs.
Response: We sincerely appreciate the reviewers advice and have made the suggested adjustment. The text in Chapters 1 and 2 has been divided into clear, readable paragraphs.
Comment 4: Please pay attention to the appropriate form of citations in the text and correct them.
Response: Thank you for this comment. All citations have been checked and corrected to ensure proper formatting.
Comment 5: Why is more than half of the cited literature older than 10 years, and many of them date back even further? Please update the References.
Response: Accurate information on biomass production of the studied species is unfortunately reported mainly in older literature. In addition, some of these older sources provide foundational information on general principles of cover crop cultivation, which is still relevant. More recent studies often cite these earlier works. We have also included up-to-date references where available.
Materials and Methods
Comment 1: The list of tested plants (line 124-127) should be included in the Material and Methods section.
Response: We thank reviewer for this helpful insight and the list of tested plants has been transferred to the MaM section.
Comment 2: There is a lack of information on the content of macro and micronutrients in soil. Enriching soil with these nutrients through intercrops is not the subject of research, but in the reviewer's opinion, a brief description and analysis of these nutrients would be desirable for agricultural research.
Response: We appreciate the reviewer’s valuable feedback and have applied the recommended change. The catch crop stands received no fertilization either before establishment of during the growing season. Their nutrient supply depended entirely on the residual effects of the preceding crop. This information is already stated in the text (L. 174).
Comment 3: The sampling method is unclear. Please describe it in more detail.
Response: Thank you, the sampling method was described in more detail:
„At each assessment location, three individual plants, including their root systems, were collected from the left side of the plot (relative to the centre) during the first assessment date, resulting in a total of 12 plants per site. Plants were carefully excavated with the surrounding soil intact to a depth of 0.2 m to preserve root structure. After excavation, roots were gently washed to remove adhering soil without damaging fine root structures, ensuring accurate measurement of underground biomass.“
Comment 4: There is no need to constantly repeat information about the years in which the experiments were conducted.
Response: We are grateful to the reviewer for this constructive remark and have implemented the recommendation. The text has been refined to avoid unnecessary repetition of experimental years.
Comment 5: Please specify what stage of plant development is indicated by the given BBCH scale grades (17-19 and 63-67).
Response: We thank the reviewer for the comment. However, we believe it si not necessary to include a detailed explanation in the text, as this information is widely available on an international scale and can be easily accessed online.
BBCH 17-19 corresponds to the leaf development stage, during which the plant has 7, 8 and 9 true leaves. BBCH 63-67 represents the flowering stages: BBCH 63 indicates that 30% of flowers have opened, 65 signifies full flowering with 50% of flowers open and 67 signifies the end of flowering.
Comment 6: Figures and tables should appear as close as possible to the point where they are first mentioned in the text. Please place Tables 1 and 2 in their appropriate places in Section 2.
Response:
We thank the reviewer for this suggestion. Tables 1 and 2 have been relocated to appear within Section 2, close to the point where they are first mentioned, as recommended.
Comment 7: Statistical methods used and the research results definitely require more detailed description.
Response: We are grateful to the reviewer for the constructive suggestion and have incorporated it: „Statistical analyses were performed using Statgraphic® Plus software, version 4.0. Differences among treatments were assessed using analysis of variance (ANOVA), followed by Tukey´s post-hoc test to determine pairwise differences at a significance level of p<0.05. Additionally, regression analyses were conducted using linear models to evaluate relationships between key variables, such as biomass production and sowing date. Data were checked for normality and homogeneity of variance prior to analysis to ensure the validity of the results.“
Results
Comment 1: Please place tables and figures in the appropriate place in the text. Please ensure that all tables and figures are referenced in the text.
Response: We thank the reviewer for the suggestion. All tables and figures have been relocated to appropriate positions in the text and are now properly cited.
Comment 2: In the reviewer's opinion, it is also worthwhile to present the results regarding carbon fixation graphically.
Response: We appreciate the reviewer’s suggestion regarding the graphical presentation of carbon fixation results. However, we have decided not to include these additional data in the manuscript, as it is already quite long and the primary focus of the study is on biomass production and allocation rather than carbon fixation. We believe that including these data would extend beyond the main objectives of the paper, but we acknowledge their potential relevance for future work.
Comment 3: The results of the statistical analyses also require more detailed description and should be appropriately summarized/analysed.
Response: We thank the reviewer for the comment. The statistical results section has been carefully expanded and clarified to improve clarity and comprehensiveness.
Comment 4: Statements such as: "Differences in the proportion of leaves on the plant are also described by Brant et al (2022) [21], (...)" - lines 275, 276, or: "The mentioned differences (...), as pointed out by Brant et al. (...)" - lines 286-289, constitute a discussion and should be moved to chapter 4.
Response: We have accepted the comment of reviewer. These statements have been moved to the Discussion section.
Comment 5: Tables 3, 4, and 5 contain a lot of information (both biomass yield results and statistical analysis results) and are therefore quite difficult to read. Please consider presenting the results and analyses more clearly.
Response: Thank you for this valuable suggestion. We recognize that these tables contain substantial information. However, we consider the current format useful for allowing direct comparison among all variables and observations.
Discussion
Comment 1: Please see note 4 in the Results section or consider combining the Results and Discussion sections into one coherent whole.
Response: We sincerely thank the reviewer for the suggestion. The manuscript has been revised to improve clarity and structure of both the Results and Discussion sections.
Conclusions
Comment 1: There is no section devoted to results in the manuscript! The summary should present substantive and synthetic conclusions based on the obtained research results, as well as discussions, if any.
Response: We thank the reviewer for this valuable comment. A dedicated Conclusion paragraph has been added, summarizing the main findings and implications of the study.
Reviewer 5 Report
Comments and Suggestions for Authors
This is an interesting and well written paper. The greatest weakness is the choice of the sewing dates. The possibility of the comparison of the first two dates with the last one is limited since the last sewing date is as cover crop whereas the first two dates prinicipally lead to seed production. It would have been more useful to compare sewing dates from Juli to September.
Some details:
Better in the title: non traditional and normally used intercrops.
Define catch crops and cover crops in the intro.
Define biomass efficiency.
Mat. and Meth.: Describe briefly the determination of organic matter in soil. No fertilization to catch crops means that soil nutrient availability especc. that of nitrogen may very important for the biomass production. Give details an the collection of hte roots.
Results: Fig. 2: Some of the results are caused by the chosen sowing dates, vegetative vs. generative growth. Fig. 4, 5 Give R2 not R. Table 1: Seeds pre m2?
Author Response
Comments and Suggestions for Authors
Comment 1: This is an interesting and well written paper. The greatest weakness is the choice of the sewing dates. The possibility of the comparison of the first two dates with the last one is limited since the last sewing date is as cover crop whereas the first two dates principally lead to seed production. It would have been more useful to compare sewing dates from Juli to September.
Response: We sincerely thank the reviewer for the insightful comments and for recognizing the value of our study. Regarding the concern about sowing dates, we would like to clarify that the selected dates are appropriate for establishing vegetative soil cover prior to the sowing the main crops such as maize, sunflower and soybean. The aim of these sowing dates is to produce an early cover crop stand that is terminated either mechanically or chemically before or shortly after the main crop is sown. This practice is standard in the Czech Republic, Germany, Austria and other countries, and under these management conditions, seed formation does not occur. Therefore, the chosen sowing dates accurately reflect typical agricultural practices for cover crop establishment in Central Europe.
Some details:
Comment 2: Better in the title: non traditional and normally used intercrops.
Response: We thank the reviewer for the suggestion. The manuscript title has been refined to: „The effect of sowing date on the biomass production on the non-traditional and commonly used intercrops from the Brassicaceae family.“
Comment 3: Define catch crops and cover crops in the intro.
Response: Thank you for raising this point, now there is new paragraph in the introduction section: Catch crops and cover crops are plant species grown primarily to protect and improve soil between main crop cycles. Catch crops are usually sown to capture residual nutrients and reduce leaching, while cover crops provide soil cover to prevent erosion, enhance soil structure and contribute organic matter and carbon sequestration.
Comment 4: Define biomass efficiency.
Response: Thank you for reviewing these terms. The term „biomass efficiency“ is not appropriately used in this context. We have revised the sentence to improve clarity and accuracy.
Early fall biomass ranged from 3 to 5.5 t/ha and was related to growing degree days (GDD) according to a logistic function [27]. Słomka and Oliveira (2021) [14] reported that the aboveground dry biomass if S. alba ranged from 0.6 to 0.8 t/ha under the studied conditions.
Comment 5: Mat. and Meth.: Describe briefly the determination of organic matter in soil. No fertilization to catch crops means that soil nutrient availability species that of nitrogen may be very important for the biomass production. Give details and the collection of the roots.
Response:
Soil organic matter content was determined using the loss-on-ignition method. Soil pH was measured potentiometrically after preparing a soil-distilled water suspension at the standard ratio.
Comment 6: Results: Fig. 2: Some of the results are caused by the chosen sowing dates, vegetative vs. generative growth. Fig. 4, 5 Give R2 not R. Table 1: Seeds pre m2?
Response: We thank the reviewer for the comment. In the graphs, R is presented because linear regression was used. In Table 1, the number of seeds per m2 is reported to determine the sowing rate as the thousand-seed weight varies between years. Consequently, the same number of seeds was sown each year to ensure consistent stand establishment.
Round 2
Reviewer 3 Report
Comments and Suggestions for Authors
The author responded well to the comments and improved the manuscript.
Author Response
Dear Reviewer,
Thank you very much for your positive feedback and for your time and effort in evaluating our revised manuscript. We sincerely appreciate your constructive comments during the review process, which have helped us improve the quality and clarity of our work. We are pleased to hear that you are satisfied with the revisions.
Thank you again for your valuable contribution.
Sincerely,
Katerina Hamouzova, corresponding author
Reviewer 4 Report
Comments and Suggestions for Authors
I would like to thank the Authors for their work in correcting the manuscript.
Please note a few additional minor comments below:
- The reviewer saw information that the crop was not additionally fertilized. This refers to the existing nutrient content in the soil. Determining this would be valuable for research. If such tests haven't been conducted, I strongly recommend considering them in the future.
- Line 62, 68, etc.: S. alba, C. abyssinica. The generic abbreviation is used when listing consecutive species of the genus. This notation may be confusing for readers unfamiliar with the genera and species of the Brasicaceae family.
Author Response
Dear Reviewer,
Comment 1:
The reviewer saw information that the crop was not additionally fertilized. This refers to the existing nutrient content in the soil. Determining this would be valuable for research. If such tests haven't been conducted, I strongly recommend considering them in the future.
Reply: Thank you very much for your helpful comment. We agree that information on the existing nutrient content of the soil is important for interpreting the results. We would like to clarify that the soil nutrient analysis had indeed been conducted prior to the experiment. These data have now been included and are provided in the Supplementary Material as requested.
We appreciate your suggestion and thank you for highlighting the importance of documenting baseline soil nutrient status.
Comment 2: Line 62, 68, etc.: S. alba, C. abyssinica. The generic abbreviation is used when listing consecutive species of the genus. This notation may be confusing for readers unfamiliar with the genera and species of the Brasicaceae family.
Reply 2:
Thank you for this helpful observation. We agree that the use of generic abbreviations may cause confusion for readers who are not familiar with the Brassicaceae genera. To improve clarity, the full Latin names of the species have now been provided at their first mention in the manuscript. Subsequent references follow standard abbreviation practices.
We appreciate your attention to detail and believe this revision improves the readability of the text.